# The Various Ways in Which Birds Blink

**DOI:** 10.3390/ani13233656

**Published:** 2023-11-26

**Authors:** John G. L. Morris, Jesse J. Parsons

**Affiliations:** 1Department of Neurology, Westmead Hospital, Sydney 2145, Australia; 2Independent Researcher, Sydney 2037, Australia

**Keywords:** nictitating membrane, eyelids, birds, avian, upper lid, lower lid, crocodiles

## Abstract

**Simple Summary:**

While our blink simply involves lowering the upper eyelid, with little involvement of the lower lid, birds have various ways of blinking depending on the species and the circumstances. Like us, they have upper and lower eyelids, though the upper lid only moves down in parrots, owls, pigeons, and a few others. The lower lid rises with drowsiness and when the bird is preening. Birds blink when they turn their heads using a ‘third eyelid’, the nictitating membrane which moves horizontally from the inner part to the outer part of the eye. The closest cousins of birds, crocodiles, blink with their lower lids and nictitating membranes but not their upper lids. As crocodiles have changed very little since they first appeared, it seems likely that their way of blinking is similar to that of the common ancestor of birds and crocodiles, the archosaur. So, upper lid blinking in birds probably came later than lower eyelid and nictitating membrane blinking. The orders of bird which have upper lid blinking are not closely related, so this type of blink is likely to be an example of convergent evolution where distantly related species evolve similar traits to adapt to similar needs. Blinking evolved as a way of keeping the cornea moist when fish first left the sea and is an as yet neglected marker of vertebrate evolution.

**Abstract:**

There has, to date, been no systematic study of the various ways in which birds blink. Digital video recordings were made, and studied using still frames, of 524 bird species, mainly in zoos but also in the wild. Videos on 106 species from various sites on the internet were studied, some of which we had also videoed, giving a total of 591 (out of a possible 10,000) species from all 43 orders and 125 (out of a possible 249) families. Digital video recordings were also made of 15 (out of a possible 24) species of crocodile. Three types of blink were observed in birds: (1) Nictitating membrane blinks were rapid and brief (phasic) and occurred mainly on head movement. (2) Upper lid blinks were seen in parrots, owls, pigeons and some others. These were also rapid and brief and accompanied nictitating membrane blinks. (3) Lower lid blinks were slow and sustained (tonic) and occurred with drowsiness and preening. Nictitating membrane blinks and lower lid blinks were seen in crocodiles but not upper lid blinks. Globe retraction, where the eyeball is pulled into the orbit of the skull during a blink, was seen in crocodiles but not birds. Phasic blinks remove debris and moisten the cornea, essential for allowing oxygen to diffuse into the cornea, which has no blood supply. Tonic blinks are probably mainly protective. The orders of birds which have upper lid blinking are not closely related and this feature is probably the result of convergent evolution.

## 1. Introduction

Blinking is of importance in an evolutionary context as it was one of the key factors which allowed fish to leave the sea and assume a terrestrial lifestyle. Most fish do not blink. They have no need to, for oxygen dissolved in water can diffuse into the epithelium on the surface of the cornea. The cornea is transparent and contains no blood vessels. Blood vessels would pose a barrier to light reaching the retina. Fish venturing from water onto land faced the problem that when the cornea dries out, atmospheric oxygen is no longer able to diffuse into the cornea (this is why the corneas of unconscious, unblinking patients in intensive care units swell and become opaque if moisture is not frequently applied to their eyes). Blinking, sweeping moisture over the cornea with eyelids, allows atmospheric oxygen to dissolve in the fluid and diffuse into the cornea.

In mudskippers, fish which spend much of their time out of water, sea water in a cup below the stalk on which the eye sits moistens the globes when the eye retracts [1]. In longer established terrestrial vertebrates, lacrimal and Harderian glands provide tears. In birds, tears are mainly produced by the Harderian gland [2] and reach the cornea through a duct which opens into the base of the nictitating membrane. As well as moistening the cornea, tears from the Harderian gland contain antibodies which protect against infection. In humans, cats and dogs, where three layers in the tear film are described, an outer layer of mucin slows the rate of evaporation of tears and lubricates the movement of the eyelids [3,4].

Video has proved to be a useful clinical tool for observing gait, eye movement and involuntary movements [5,6] and was used in this study to investigate how birds blink. Blinking in humans can be a reflex (for example, the corneal reflex), voluntary or spontaneous [7]. While spontaneous blinking in humans is a relatively simple process, the upper lid briefly falling and rising about twelve times a minute [8], birds have more than one type of spontaneous blink, and not all species blink in the same way.

In one of the earliest accounts, Owen [9] noted that, like non-avian reptiles, birds have three eyelids. The upper and lower lids move in the vertical plane. The third, the nictitating membrane, is drawn horizontally or obliquely across the cornea by the combined action of the *quadratus membranae nictitantis* and *pyramidalis membranae nictitantis* muscles which lie behind the eyeball [10].

Owen [9] observed that the lower eyelid rises in sleep. Owls and nightjars blink with their upper lids. The upper eyelid is elevated by *levator palpebrae superioris* and the upper and lower eyelids are brought together by the action of *orbicularis oculi*. In an anatomical study, Stibbe [11] contrasted the nictitating membrane blink of birds with that of mammals. In the latter, the membrane is not actively drawn across the eye by the contraction of muscles. Rather, it lies in a state of elastic tension out of sight on the medial side of the globe, springing across when the eyeball retracts by the action of the *retractor bulbi* muscle. In birds, the muscles required for globe retraction, and in most birds for eye movements, have become vestigial—capable of causing only low-amplitude eye movement of the type, for example, which causes saccadic oscillation [12]. This evolved presumably to lighten the load for flight.

In cats and dogs, globe retraction only occurs during blinking with both eyelids, so the membrane is not usually seen. Blount [13] measured the inter-blink interval in a wide range of mammals, some reptiles and amphibia, and in four species of birds. In an eagle owl, often the blinks involving the eyelids or nictitating membrane were unilateral. Of relevance to the present study, it was noted that in a parrot, the lower lid was much slower to return to its resting position than the upper lid. Most fish do not have functioning eyelids and Blount surmised that blinking evolved in air-breathing animals as a means of preventing desiccation of the cornea by exposure to air. Mowrer [14] proposed that blinking during head movements, which are rapid and frequent in birds as they have to compensate for their lack of eye movements, might prevent blurred images reaching the retina. During rapid head turns, images traverse the retina at a rate beyond the capacity of the retinal photoreceptors to respond adequately [15]. Yorzinski [16] has more recently proposed that blinking during a gaze shift, a time when visual information is already limited, may minimise additional information loss. Kirsten and Kirsten [17] measured the spontaneous blink rates of 25 species of birds. They also noted a correlation between blinks and head turns. Nocturnal species such as owls had a lower blink rate than diurnal species. Curio [18] coined the terms ‘tonic’ and ‘phasic’ blinks to distinguish slow prolonged blinks from rapid brief ones. In the tonic blinks of many orders (Anseriformes, Accipitriformes, Falconiformes, Galliformes, Charadriiformes, Columbiformes and the Oscines suborder of Passeriformes), the lower lid rises during sleep. In Psittaciformes and Trochili (hummingbirds), the upper lid descends in sleep and in Strigiformes and Caprimulgiformes, both lids are involved. He commented that ‘such information is lacking for most orders, or the handbooks provide wrong or conflicting information’.

The purpose of the present study was to take up Curio’s challenge and increase our understanding of the way that birds blink across the avian orders. To do this, we aimed to video or obtain videos of representative numbers of species from every order of birds to test the following hypotheses: Do all birds have the same type of blink? Is the blink type related to activities such as head turns, drowsiness, nocturnality, preening or ground feeding/pecking? Where a blink type is confined to certain species of bird, can this be related to phylogeny? How does blinking in birds compare with blinking in their closest cousins, Crocodilia?

The following predictions were made: Such has been the evolutionary flowering of birds and the myriad environments to which they have adapted, not all birds will have the same types of blink. Blinking in birds will differ from blinking in crocodilians, which have changed little since they first emerged.

## 2. Materials and Methods

Birds were videoed throughout the year from 2015–2021 using a Panasonic Lumix DMC FZ2500 digital single-lens reflex camera and a Sony FDR–AX53 camcorder. The zoom on these cameras allowed close-up views of the eyes even when the bird was more than 5 metres away. There was no intervention such as making a noise, gesticulating or using flash photography. Most filming was carried out at 25 frames/second and in some settings, for example, to see saccadic oscillation, at 100 frames/second. If they remained in view, birds were videoed for at least a minute. When the opportunity arose, the same bird might be videoed repeatedly. Wild birds were studied in Sydney, Australia, and to a small extent in the UK and the Netherlands. Captive birds and other animals were filmed at zoos in Australia: Taronga and Featherdale Park in Sydney, the Reptile Park in Gosford and zoos in Canberra, Melbourne, Adelaide, Launceston and Queensland (Australia Zoo, Lone Pine and Currumbin); Singapore: the Jurong Bird Park; UK: London Zoo, Whipsnade, Barnes Wetlands, Torquay, Woburn Abbey, Bristol, Crocodiles of the World in Carterton, Oxfordshire and the Suffolk Owl Sanctuary; Ireland: Dublin; Belgium: Pairi Daiza Zoo; France: ZooParc de Beauval; Netherlands: Rotterdam Zoo, the Avifauna Park; USA: Bronx Zoo in New York, San Diego Zoo; South Africa: Pretoria, Hout Bay and Stellenbosch; New Zealand: Auckland; Greece: Athens. Sampling was opportunistic depending on which species the zoos happened to have. Common species were often videoed multiple times and no major differences were ever observed between blinks in different individuals of the same species. Rare species might be videoed only once. 

Digital video files were analysed using Movavi Video Editing software [Versions 12.1 to 15.4.1] (which is easy to use and not overly expensive), made in Novosibirsk, Russia, with a Dell laptop (XPS 15 9560), made in Longhua, China, and later a MacBook Pro (2.6 GHz Processor and 16 GB memory), US Apple Company, manufactured in China. Video files could be viewed frame by frame in order to follow the position of the eyelids and nictitating membrane during the course of a blink. The software has a facility for taking still pictures. Examples are shown in Figure 1, Figure 2, Figure 3, Figure 4, Figure 5, Figure 6, Figure 7, Figure 8, Figure 9, Figure 10, Figure 11, Figure 12, Figure 13, Figure 14, Figure 15 and Figure 16. In order to obtain a more complete representation across the orders, videos from a further 106 species from the internet were studied. These were played back at a quarter speed, providing clear information on the positions of the eyelids and nictitating membrane during the course of the blink. Still photos were not taken. The identities of species were confirmed using the Cornell University site Birds of the World (https://birdsoftheworld.org).

Videoing birds was not without its challenges. Stationary birds tended to preen themselves, obscuring the eyes, or to fall asleep. With birds with laterally situated eyes, usually only one eye could be seen at a time. This was not the case with the frontally oriented eyes of raptors. Nictitating membrane blinks were mainly associated with head movements and in small birds, such as finches (*Fringillidae*), honeyeaters (*Meliphagidae*) and fairywrens (*Maluridae*), these were so rapid that any blinking which might have occurred was often obscured by movement blur. It was sometimes hard to detect the movement of transparent nictitating membranes, particularly across a dark iris. This was the case in a number of species of the order Anseriformes. For the purposes of this study, no time limit was set on how long a blink could last, but for most species, blinks rarely lasted longer than a second or two, and most were much briefer.

## 3. Results

### 3.1. Birds

Video recordings were made on 591 bird species, which included 43 orders and 125 families (see Table A1). No blinking occurred during the period of observation (i.e., the bird flew away) in 67 species. Three types of blink were observed:

### 3.2. Nictitating Membrane Blinks

Here, the membrane travelled across the cornea from the inner (medial) to the outer (lateral) canthus (Figure 1).

Nictitating membrane blinks were usually frequent, rapid, brief and coincided with head movement. They also occurred with the head still. In the few species with preserved eye movements, nictitating membrane blinks also coincided with these (Figure 2).

Eye-movement-related nictitating membrane blinks were seen in Bucerotiformes and some species of Anseriformes, Ciconiiformes, Cuculiformes, Otidiformes, Passeriformes and Sphenisciformes. Eye movements were most noticeable in birds with large, long bills like hornbills (Bucerotidae), storks (Ciconiidae) and pelicans (Pelicanidae) but also in parrots (Psittaciformes) with large, short bills.

Nictitating membrane blinks were seen in the following orders: Accipitriformes, Anseriformes, Bucerotiformes, Cariamiformes, Casuariiformes, Cathartiformes, Ciconiiformes, Coraciiformes, Cuculiformes, Eurypygiformes, Falconiformes, Galliformes, Gruiformes, Musophagiformes, Otidiformes, Pelecaniformes, Phoenicopteriformes, Piciformes, Sphenisciformes, Suliformes and Tinamiformes.

In some species, the nictitating membrane blink was accompanied by a partial upper lid blink with slight lowering of the medial (nasal) side of the upper eyelid (Figure 3). In these species, the upper lid never covered the pupil. Partial upper lid blinks were seen in many species of Ciconiiformes, Galliformes and Pelecaniformes and a few species of Accipitriformes, Charadriiformes and Musophagiformes.

Birds with minimal eye movements, and even in those with easily observable eye movements, surveyed the landscape around them by frequent, rapid, brief head movements, akin to the saccadic (rapid, fixed velocity, voluntary) eye movements of humans [19]. Most, but not all, head movements were accompanied by nictitating membrane blinks.

Nictitating membrane blinks could be unilateral (‘winks’) or bilateral—asynchronous or synchronous. In larger species with less opaque membranes, arcades of blood vessels became visible during the course of a blink (Figure 4).

Nictitating membrane blinks were observed in 376 of 524 species which blinked (Table A1, and Table 1) and could not be seen in 91 species which had a blink of another sort (in some of these cases, the quality of the video was good enough to see eyelid movement but not nictitating membrane movement). In 70 species, the membrane was transparent. This was the case in 21/22 species of Anseriformes, 3/3 species of Phoenicopteriformes, 3/3 of the family Spheniscidae (order Sphenisciformes), 5/5 of Suliformes and in some species within Charadriformes, Columbiformes, Galliformes, Pelecaniformes and Strigiformes. While most of the species with transparent nictitating membrane were aquatic, there were many notable exceptions, including owls (Strigiformes). In 9/13 species of Bucerotiformes, nictitating membrane blinks were associated with eye movements.

More prolonged nictitating membrane blinks were associated with irregular rapid bobbing of the eye. This was described by Pettigrew [12] as saccadic oscillation. The shaking of the pecten, a highly vascular structure protruding into the vitreous humour, increases the release of oxygen and nutrients into the vitreous humour. These diffuse into the retina which, unlike the human eye, does not have its own blood supply [20]. Saccadic oscillation was seen most clearly in birds with large eyes such as the stone curlew. It was not seen with the naked eye—it requires 100 fps video capture.

### 3.3. Upper Eyelid Blinks

Here, there was downward (ventral) movement of the upper eyelid accompanied by horizontal movement of the nictitating membrane from the inner (medial) to outer (lateral) canthus (Figure 5), although the upper eyelid often obscured this (Figure 6).

As with ‘pure’ nictitating membrane blinks, upper lid blinks were usually rapid, brief and coincided with head movement. Occasionally, they were observed with the head still. The movements of the upper lid and nictitating membrane appeared to be linked during a blink, neither occurring alone, but their movements were not necessarily synchronous, one starting or finishing before or after the other.

Upper lid blinking was observed in 166 of 524 species which blinked (Table 1 and Table A1) and in 43 of these, accompanying movement of the nictitating membrane could be seen. Upper lid blinking was seen in 30/36 species of Columbiformes, in all 60 species of Psittaciformes, all 11 species of Caprimulgiformes and in 24/28 species of Strigiformes. In other orders, the proportion of upper blinks was lower: 7/40 species of Charadriiformes and 18/131 species of Passeriformes (8/14 of the family Estrildidae). In no case did a nictitating membrane blink occur without an accompanying upper lid blink in species displaying upper lid blinks.

In some species, the upper lid was made prominent by being a different colour from the rest of the head e.g., *Caloenas nicobarica* and *Momotus coeruliceps,* which have white upper lids.

### 3.4. Lower Eyelid Blinks

A total of 77 species (out of 524 species which blinked, Table 1 and Table A1) across most orders of bird were observed while preening (Figure 7) and 98 during drowsiness (Figure 8). These had lower lid blinks in which the lower lid rose, usually slowly and in a sustained manner. Before sleep, the lower lid would rise and fall repeatedly, before eventually remaining elevated. Often, the nictitating membrane could be seen making a pass as the lower lid rose (Figure 9), but it did not usually stay out for the entire duration of the lower lid blink, unlike upper lid blinks.

The lower lid in some species (e.g., *Radjah radjah*, *Columba guinea*) was paper thin, presumably allowing some light to enter the pupil. In others it was of a different colour to the rest of the head, making it prominent (e.g., *Hypotaenidia philippensis*). Rarely, species used the nictitating membrane rather than the lower lid during preening (e.g., *Scopus umbrette*).

In addition to the three types of blink, two further behaviours involving blinking were noted:

### 3.5. Pecking

Domestic hens (*Gallus gallus domesticus*) were observed pecking lichen off an old corrugated-iron panel. Their bills had been worn down. What was noticeable was that with every peck, a nictitating membrane blink occurred as the bill reached a certain distance from the target. In pigeons (Columbiformes), each peck was accompanied by an upper and lower lid blink (Figure 10). The type of blink seen with pecking varied with the species:

Lower lid blink on pecking: *Numenius madagascariensis*, *Alectoris chukar*, *Phasianus colchicus*, *Colinus virginianus*, *Nymphicus hollandicus*.

Upper lid blink on pecking: *Turnix melanogaster*, *Geopelia cuneata*, *Geopelia striata*, *Ocyphaps lophotes*, *Spilopelia chinensis*, *Spilopelia senegalensis*, *Streptopelia risorii*, *Nymphicus hollandicus*.

Upper and lower lids involved in pecking: *Pluvialis fulva*, *Columba livia domestica*, *Columba palumbus*, *Coturnix japonica*, *Taeniopygia guttata*, *Barnardius zonarius*, *Platycercus eximius*, *Psephotus haematonotus*.

### 3.6. Synergic Upper and Lower Eyelid Moment

A curious phenomenon seen mainly in Columbiformes, Psittaciformes and Strigiformes in states of drowsiness involved the upper lid slowly descending until it reached the lower lid in its resting position. The lower lid would then slowly rise, the upper lid rising at the same pace so that the eye remained closed. The two lids would then rise and fall together in unison, until the eye eventually opened (Figure 11).

Sometimes, the bird would squint, the lids coming together leaving a small gap or slit through which the pupil could be seen (Figure 12, Figure 13 and Figure 14)

Sometimes, the nictitating membrane was visible during a squint (Figure 15).

### 3.7. Crocodilia

Videos were made of 15 species of Crocodilia from 3 families (Table A2). This was a more time-consuming exercise than with birds, as many minutes could pass between blinks. Nictitating membrane blinks were seen in 15 species, 3 with visible globe retraction, 4 with no evident globe retraction (Figure 16) and 11 associated with globe retraction and elevation of the lower lid (Figure 17). In five species, there was lower lid elevation and globe retraction without visible movement of the nictitating membrane. In no case was there upper lid blinking, though in some cases the upper lid sank a little into the orbit as the globe retracted.

**Figure 16 animals-13-03656-f016:**
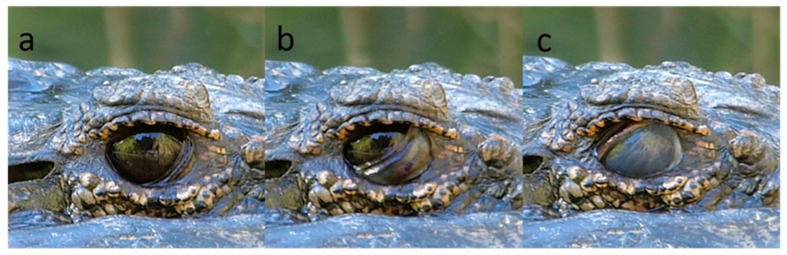
Nictitating membrane blink in an American Alligator (*Alligator mississippiensis*). Sequence: (**a**): pre-blink, edge of nictitating membrane visible at inner fornix; (**b**): nictitating membrane half covers the cornea; (**c**): nictitating membrane covers the cornea. No change in diameter of palpebral fissure, so no evidence of globe retraction.

## 4. Discussion

The present study confirms the observation of Curio [18] that blinking in birds appears to be of two types, phasic and tonic.

Phasic blinks were frequent, rapid and brief and involved the nictitating membrane and, in some species, the upper lid as well. They occurred during head movement associated with a shift in gaze [16]. Not every shift in gaze was accompanied by a blink, and some blinks, even flurries of blinks, were observed with the head still. Phasic blinks spread tears from the Harderian gland over the surface of the cornea. Debris on the surface of the cornea is also swept towards the medial canthus. Tear fluid drains into puncti on the inner end of the upper and lower eyelid and into the lacrimal duct, and thence to the anterior/cranial nasal cavity.

Tonic blinks were infrequent, slow and sustained, occurring with drowsiness and during preening and involving the nictitating membrane and lower lid. They appear to have the function of protecting the eye from feathers and ‘feather dust’ during preening (and from the bill of other birds preening them). The lower lid would rise and fall as a prelude to sleep and then remain raised during sleep. This protects the eye from exposure and foreign material and shuts out light. That said, it was not uncommon to find that the lower eyelids were paper thin when closed. Tonic blinks are likely to be less effective than phasic blinks in distributing tear fluid and removing debris as they occur less frequently.

During prolonged blinks and during sleep, the cornea is no longer exposed to atmospheric oxygen. In humans, gas exchange continues during sleep by virtue of the rich supply of capillaries on the conjunctival side of the eyelids [21]. If these are also present in birds, gas exchange might be impaired when the nictitating membrane comes between the cornea and the eyelids. It may be significant therefore that the nictitating membrane appears to have a rich arcade of blood vessels.

Involvement of the lower lid in tonic blinks is seen in frogs, turtles, lizards (personal observations), crocodiles as well as birds. This may hark back to the time when fish leaving the sea adopted an amphibious lifestyle. Half-submerged in water, elevation of the lower lid protects the eye from floating debris. In crocodiles, a ‘submergence reflex’, where the lower eyelid elevates and the external auditory meatus closes as the water level rises, has been described [22].

Of particular interest is the finding that in some orders and families of birds, phasic blinks involve not only the nictitating membrane but also the upper lid. It is likely that this feature evolved after the emergence of nictitating membrane blinks, as Crocodilia only blink with their nictitating membranes and lower lids. With birds, Crocodilia are the only living descendants of archosaurs, a group of diapsid amniotes which once included dinosaurs and pterosaurs. While birds have evolved over the last 165–150 million years into 10,000 or so extant species ranging greatly in size, shape and habitat [23], Crocodilia have evolved into only 23 extant species and have changed little since they first appeared [24]. Upper lid blinks were not found in Paleognathae (ratites and tinamous) or Galloanserae (land fowl and waterfowl), only in the Neoaves clade which underwent a rapid expansion following the Cretaceous–Paleogene mass extinction [25].

Upper eyelid blinks were seen mainly in four orders: Columbiformes, Strigiformes, Psicattiformes and Caprimulgimorphae. The question arises, what were the factors which led to this? Have they perhaps evolved from a common ancestor which blinked with its upper eyelids? According to a recent genomic analysis of clade genotypes [25], these four orders are not closely related (Table A3). Columbiformes are classified within Columbaves which includes turacos, which do not blink with their upper eyelids. Strigiformes lie between Accipitriformes and Coraciimorphae, neither of which blink with their upper eyelids. Psicattiformes are classed with Australaves, a group which includes falcons, which do not blink with their upper eyelids. Caprimulgimorphae is an order which includes swifts and hummingbirds. In short, no clear pattern of genetic association emerges. Further evidence that upper lid blinking is not related to clade genotypes comes from the finding that even within orders such as Passeriformes and Charadriiformes, which mostly blink with their nictitating membranes, there were families which blinked with their upper lids.

No clear association was apparent between upper lid blinking and nocturnality. Of the orders with upper lid blinks, Strigiformes is the only one which is almost exclusively nocturnal (the exceptions being the northern hawk-owl, *Surnia ulula*, and the burrowing owl, *Athena cunicularia*). Within the order of Caprimulgiformes, nightjars (Caprimulgidae) are nocturnal but swifts (Apodidae) and hummingbirds (Trochilidae) are not. Columbiformes and Psittaciformes are diurnal.

These findings suggest that upper lid blinking is the result of convergent evolution. So, what might have been the factors which favoured upper lid blinking? Movement of objects in air tends to be of a higher velocity than in water and therefore more potentially injurious. Birds cannot retract their eyes, and it is retraction in other classes of animal which provides the most protection from blunt mechanical injury [26]. While the lower lid can cover the eye, it tends to move slowly. The rapidly moving upper lids and nictitating membranes might provide reflex closure of the eyes in response to a visual threat (menace reflex) [27], touching of the feather-equivalent of eyelashes by potentially damaging objects like twigs or leaves (lash response) [28] or touching the cornea (corneal reflex) [7].

This begs the question, are birds which blink with their upper lids at greater risk of eye injury than other birds? This may be the case in ground-feeding birds like pigeons (Columbidae) and quails (Phasianidae) which peck for seeds in undergrowth and foliage. A common finding was of blinking with every peck but this could involve the nictitating membrane, upper lid, lower lid or both, depending on the species. Protection during blinking does not depend on having a mobile upper lid. In the case of owls and nightjars, catching live prey capable of injuring their eyes at night may put them at greater risk than other raptors which hunt during the day when visibility is better. But the lower lid and nictitating membrane could also provide this protection. A paper by Ostheim and colleagues [29] raised another possibility. The amount of light focused on the retina by the cornea and lens is related to the size of the pupil. Ostheim and colleagues [29] showed that, in the case of pigeons, the eyelids may also play a role under certain conditions. They found that during pecking, pigeons do not close their eyes completely. The effect of this is to turn the round aperture of the pupil into a narrow horizontal slit, thereby increasing the depth of field [30]. This would result in sharper retinal images of the target, even at close range. For this to occur, upper and lower lids need to come together synergistically. Such behaviour was observed in pigeons, parrots and owls, albeit with drowsiness.

It is hard to know what to make of the many species of bird with partial upper lid blinks—where the inner part of the upper lid lowered a little during a nictitating membrane blink. It did not appear to be produced by eye movement. Perhaps, in these species, upper lid blinks are in the process of evolving into existence—or perhaps out of existence?

Birds and crocodilians both blink with their nictitating membranes to lubricate and clear debris from the cornea. Crocodilia and aquatic birds, such as cormorants (Phalacrocoracidae), may protect their eyes while submerged without losing all vision, using their nictitating membranes, though this has not been established. Certainly, there is a preponderance of transparent nictitating membranes in birds which forage aquatically. Under water, the cornea and nictitating membrane contribute little to refraction of light entering the eye as the cornea is thin and the refractive indices of water and aqueous humour in the anterior chamber of the eye are similar [31]. Birds protect their eyes during sleep and preening by raising the lower eyelids, which are often quite flimsy. By contrast, Crocodilia protect their eyes from injury by retracting the eyeballs deep into the orbits. In saltwater crocodiles, this causes the thickly armored, hinged, upper eyelids to fold like a trapdoor to cover over the orbits. The lower eyelids are also raised. Having larger prey than birds, the requirements of Crocodilia for protection of the eyes are likely to be more stringent than is the case for most birds. In the evolutionary balancing act of reducing body weight while increasing visual acuity, birds have mostly abandoned extraocular muscles which move the eyes and retract them, while increasing the size of the eyes relative to their body mass.

## 5. Conclusions

Birds (and Crocodilia) have two types of blink: rapid brief (phasic) nictitating membrane blinks; and slow sustained (tonic) lower eyelid blinks. In some avian orders, and in some families within others, phasic nictitating membrane blinks are accompanied by upper eyelid blinks.

Phasic blinks occur on head turn and pecking, tonic blinks on preening and with drowsiness. Nocturnality does not particularly predispose to any blink type.

Upper lid blinking cannot readily be related to phylogeny. The orders where it occurs in every species are not closely related. This type of blinking is probably the result of convergent evolution.

## Figures and Tables

**Figure 1 animals-13-03656-f001:**
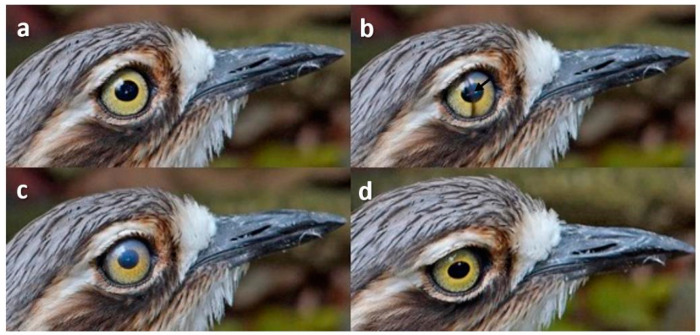
Nictitating membrane blink on head movement in a Bush Stone-Curlew (*Burhinus grallarius*). Sequence: (**a**): pre-blink; (**b**): transparent nictitating membrane has crossed from the medial canthus to halfway across the pupil; (**c**): the membrane has covered the cornea (which appears slightly opaque as a result); (**d**): post-blink.

**Figure 2 animals-13-03656-f002:**
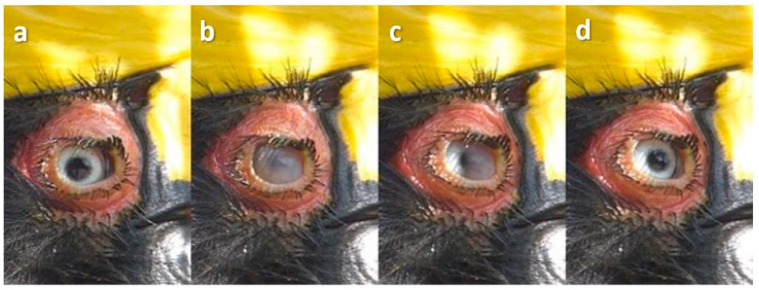
Nictitating membrane blink on eye movement in a rhinoceros hornbill (*Buceros rhinoceros*). Sequence: (**a**): pre-blink; (**b**): cloudy nictitating membrane has covered the cornea; (**c**): the membrane is moving back towards the inner canthus; (**d**): post-blink.

**Figure 3 animals-13-03656-f003:**
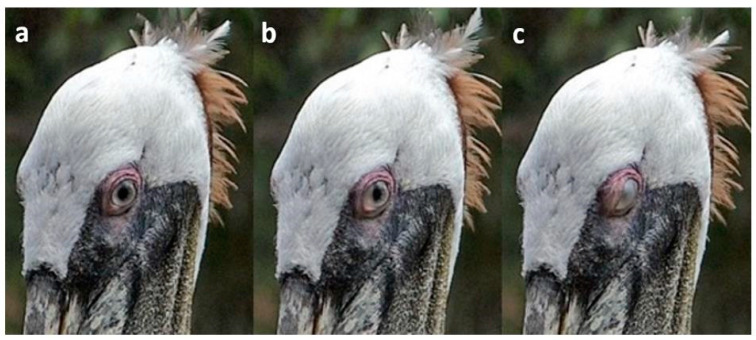
Nictitating membrane blink with partial upper lid blink in a Brown Pelican (*Pelecanus occidentalis*). Sequence: (**a**): pre-blink; (**b**): cloudy nictitating membrane has moved from the inner canthus to the edge of the pupil; (**c**): the membrane has covered the cornea and the part of the upper lid closest to the inner canthus has lowered a little.

**Figure 4 animals-13-03656-f004:**
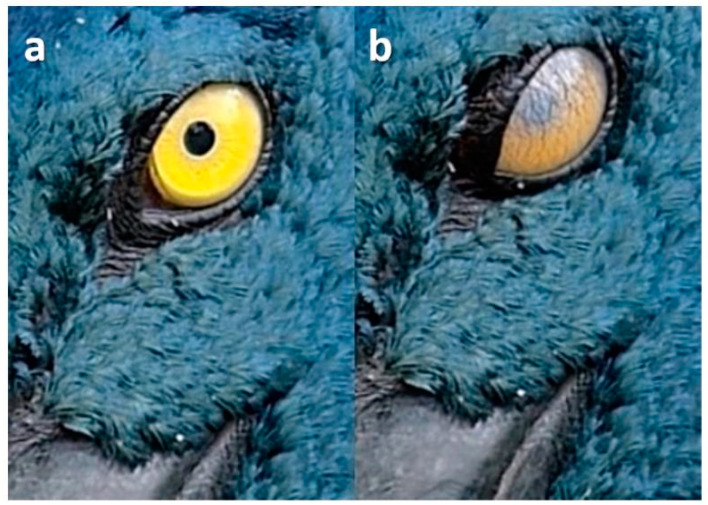
Nictitating membrane blink showing vascular arcades in a Black-Necked Stork (*Ephippiorhynchus asiaticus*). Sequence: (**a**): pre-blink; (**b**): blood vessels visible in the nictitating membrane during a full blink.

**Figure 5 animals-13-03656-f005:**
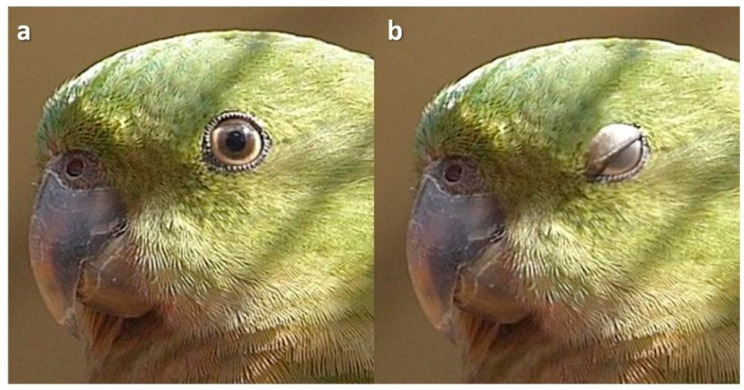
Upper lid and nictitating membrane blink in a female Australian King Parrot (*Alisterus scapularis*). Sequence: (**a**): pre-blink; (**b**): the upper lid is half lowered and a cloudy nictitating membrane is seen below it.

**Figure 6 animals-13-03656-f006:**
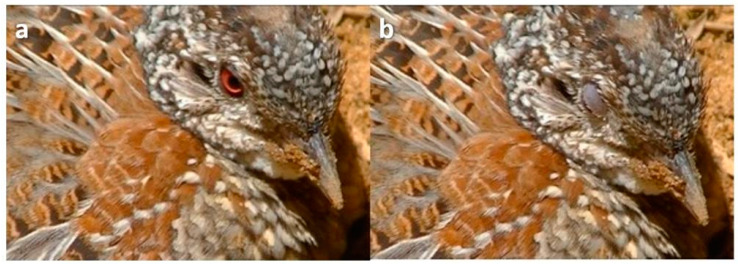
Upper lid blink in a Painted Buttonquail (*Turnix varius*). Sequence: (**a**): pre-blink; (**b**): full upper lid blink.

**Figure 7 animals-13-03656-f007:**
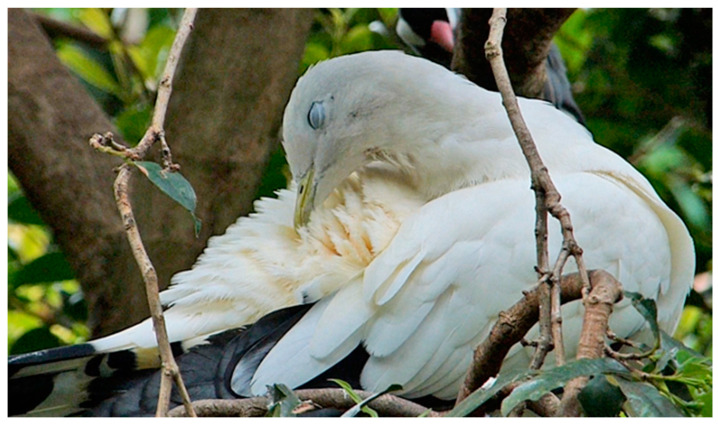
Lower lid elevation during preening in a Pied Imperial Pigeon (*Ducula bicolor*).

**Figure 8 animals-13-03656-f008:**
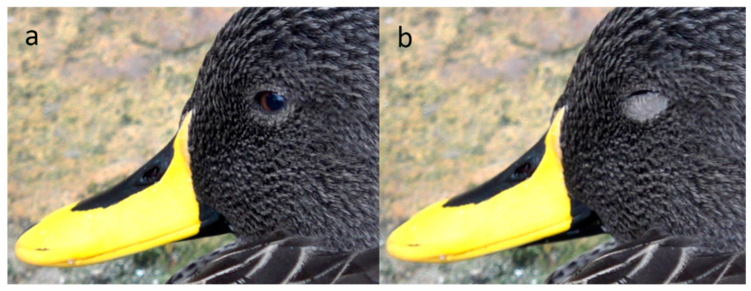
Lower lid elevation in a Yellow-Billed Duck (*Anas undulata*). Sequence: (**a**): awake; (**b**): asleep.

**Figure 9 animals-13-03656-f009:**
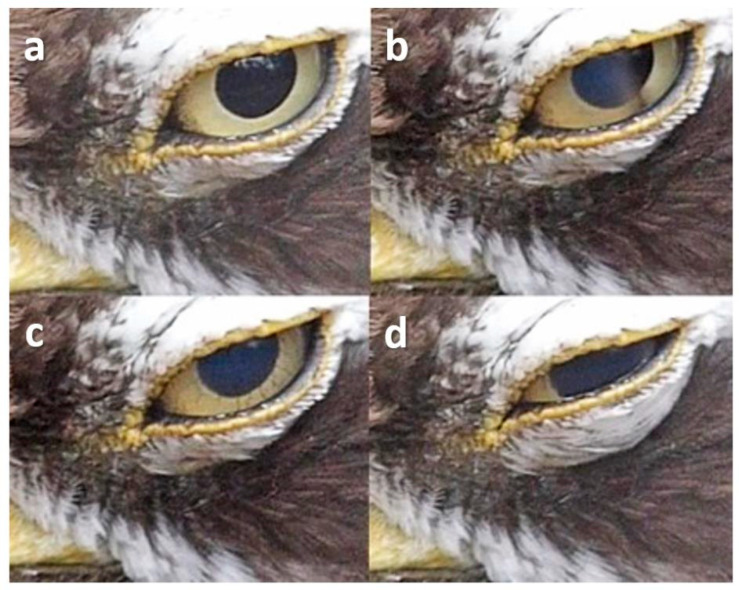
Nictitating membrane and lower lid blink in a Beach Stone-Curlew (*Esacus magnirostris*). Sequence: (**a**): pre-blink; (**b**): transparent nictitating membrane crosses the pupil from the inner canthus; (**c**): membrane has covered the cornea; (**d**): lower lid starts to elevate, and the pupil has dilated.

**Figure 10 animals-13-03656-f010:**
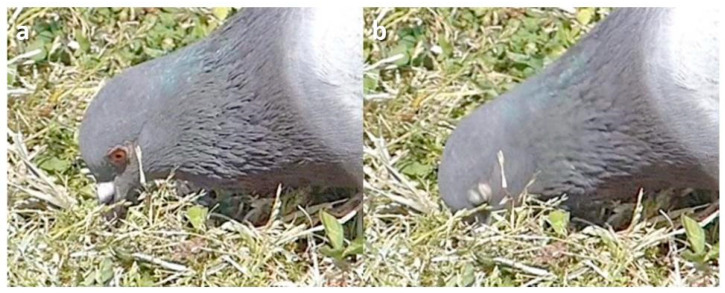
Upper and lower lid blink during pecking in a Feral Pigeon (*Columba livia domestica*). Sequence: (**a**) Pre-blink; (**b**) Upper and lower lids come together.

**Figure 11 animals-13-03656-f011:**
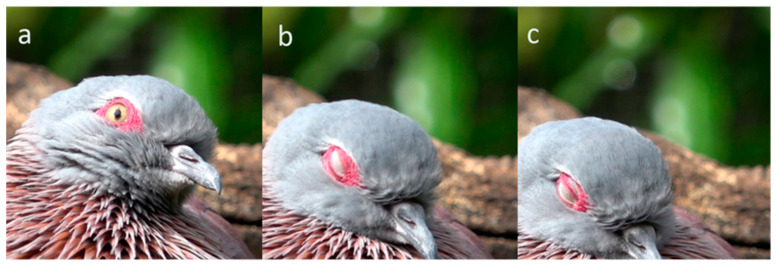
Synergic upper and lower lid blink in a Speckled Pigeon (*Columba guinea*). Sequence: (**a**): pre-blink; (**b**): full upper lid blink; (**c**): the lower lid has risen, ‘pushing’ the upper lid upwards. Both lids are pale and paper thin.

**Figure 12 animals-13-03656-f012:**
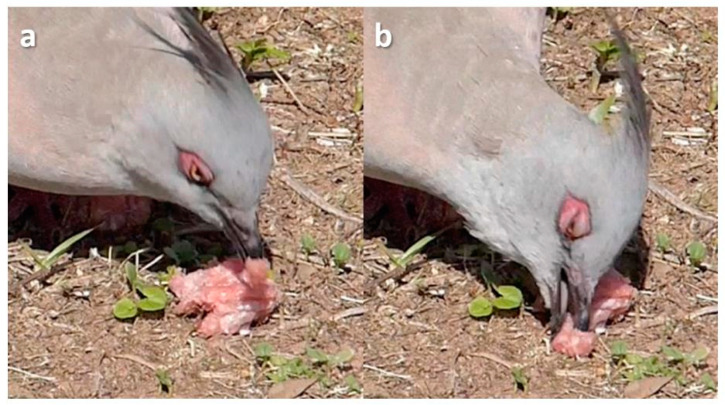
Squint in a Crested Pigeon (*Ocyphaps lophotes*) while pecking. (**a**) The upper lid starts to lower; (**b**) The lower lid rises leaving a slit through which the pupil is visible.

**Figure 13 animals-13-03656-f013:**
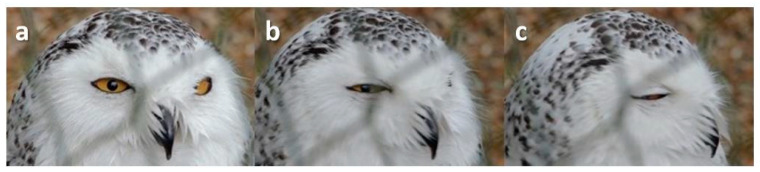
Squint on head turn in a Snowy Owl (*Bubo scandiacus*). (**a**) Pre-blink; (**b**) Both lids start to come together, the pupil remaining visible; (**c**) A small gap remains between the lids.

**Figure 14 animals-13-03656-f014:**
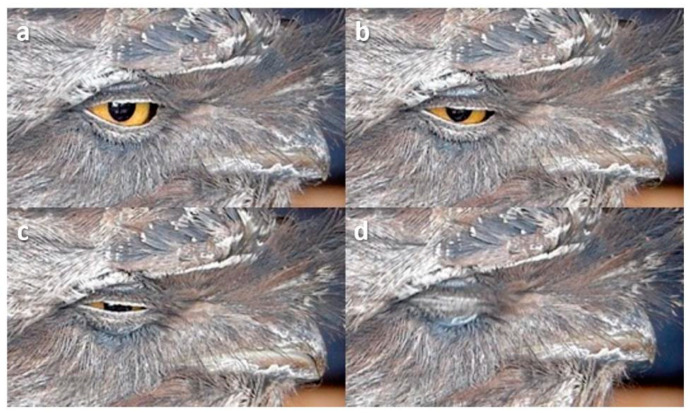
Squint in a Tawny Frogmouth (*Podargus strigoides*). (**a**) Pre-blink; (**b**) Upper lid lowers a little; (**c**) Lower lid rises but pupil remains visible; (**d**) Lids come together.

**Figure 15 animals-13-03656-f015:**
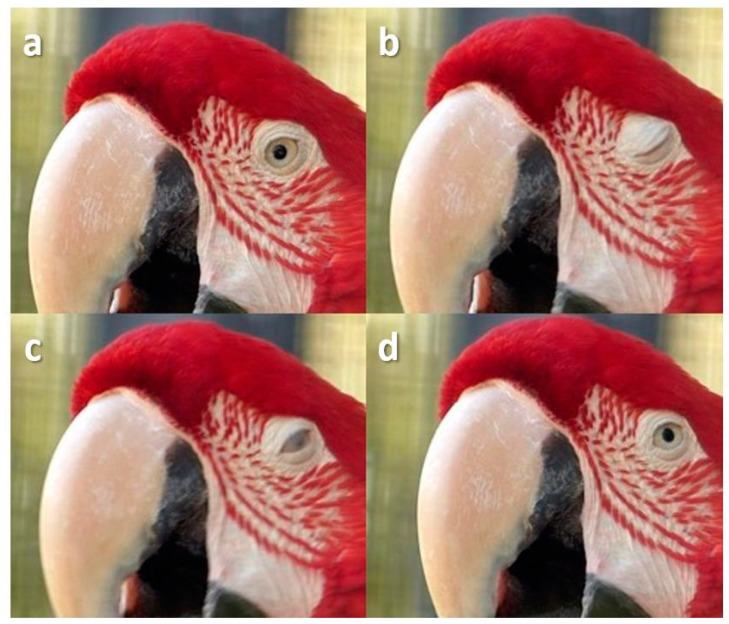
Squint in a Red-And-Green Macaw (*Ara chloropterus*) during slight head movement. (**a**) Pre-blink; (**b**) Upper lid falls and lower lid rises, causing them to come together; (**c**) Upper lid rises a little exposing the nictitating membrane; (**d**) Post-blink.

**Figure 17 animals-13-03656-f017:**
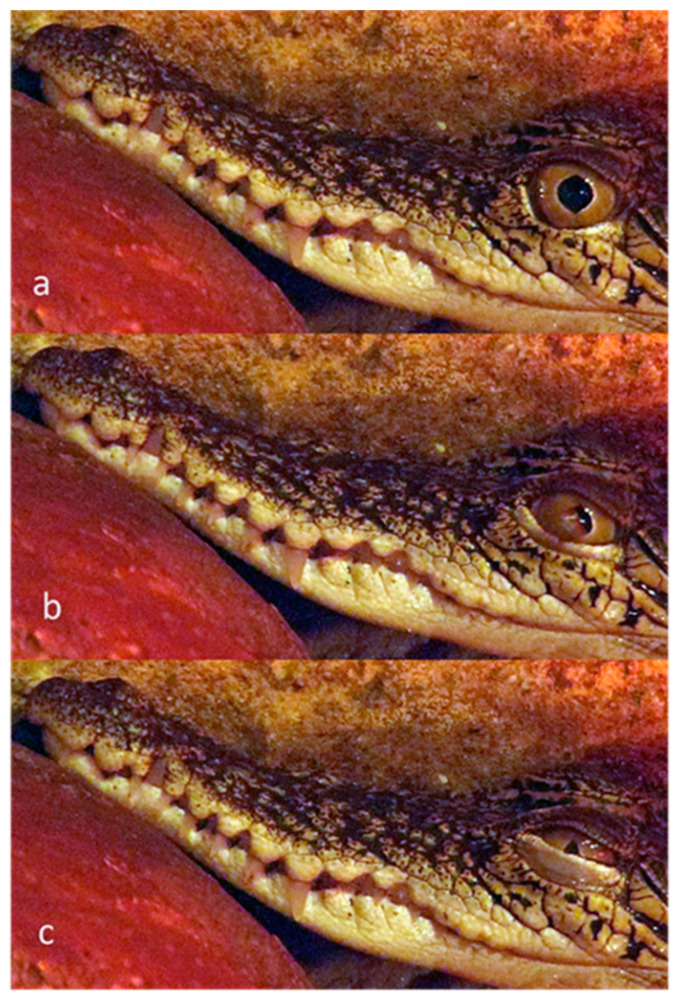
Nictitating membrane and lower lid blink in a drowsy juvenile Saltwater Crocodile (*Crocodylus porosus*). Sequence: (**a**): pre-blink; (**b**): nictitating membrane half covers the pupil and lower lid starts to rise; (**c**): lower lid half elevated.

**Table 1 animals-13-03656-t001:** Number of species with each of the three types of blink and the total number of species videoed where a blink occurred.

Blink Type	Number of Species	Number Videoed
Nictitating membrane	376	524
Upper lid	166	524
Lower lid	77	524

## Data Availability

Data are contained within the article.

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
