# Peer review of "The Various Ways in Which Birds Blink"

_animals, 2023, doi:10.3390/ani13233656_

Round 1
Reviewer 1 Report
Comments and Suggestions for Authors
I very much enjoyed reading this paper and I think that it is wonderful that an experienced neurologist has found a new interest in biology, to which I'm sure he can contribute significantly.
I would make a couple of suggestions and please find attached your paper with some of my comments on it. There are just a couple of minor grammatical points, being a touch of punctuation and a comment that I believe i.e. should have a full stop after the 'i' and the 'e', as of course it is an abbreviation for id est. I would put all genus and species names in the text in italics, and capitalise order and family names, e.g. Crocodilia; I think this is important as not italicising genus and species names is incorrect. I note that in the vast majority of the paper you use scientific names and, often, the common name alongside it. I think perhaps in those few instances where you use a common name only you should probably put the scientific name also, to be consistent, e.g. 'finches' on page 4. I don't think it's necessary for dogs and cats but perhaps consider this, to be consistent everywhere, if you feel it does not interfere with the ease of reading.
I note also that in the text you refer to the muscles by their Latin names (as per the Nomina Anatomica Avium) but in figure 1 by the anglicised form; I think this should be made consistent. I personally would prefer the Latin name. I see that you have used the English name for other anatomic structures; I would prefer the Latin name (e.g. membrana nictitantis) personally although this could be considered personal preference and certainly writing nictitating membrane is both correct and reads well.

It is clearly well written, apart from the minor points raised above.
Author Response
Thank you for your kind review. I have made all the changes you suggested: i.e.; Crocodilia, lines 347 and 512 and; full Latin names for the muscles, line 51; finches (Fringillidae); honeyeaters (Meliphagidae); fairywrens (Maluridae), lines 148-149; the Latin names for the extraocular muscles from the Handbook of Avian Anatomy in Figure 1, line 55; videoed, line 114; the Latin names for the birds in italics in Figures 2-18; extraocular for orbital on line 508; latin name for domestic hen, line 301; I have moved the references [19] and [20] to line 386 as they support the proposition that the outer layer of mucin in tears reduces the rate of evaporation.
Reviewer 2 Report
Comments and Suggestions for Authors
The study by Morris and Parsons aims to shed light on the ways in which birds blink their eyelids. To do this, the authors collected images of 591 bird species and used them to categorize the type of blink each species uses. The authors’ also collected similar images of 15 crocodilian species since this order is the closest living relatives to birds and the comparison offers inferences about the ancestral condition for archosaurs. The authors observed three types of blinks in birds (nictitating membrane blinks, upper lid blinks, and lower lid blinks). Crocodilians exhibited nictitating membrane blinks and lower lid blinks, but not upper lid blinks. This study clearly involved a monumental effort to collect and analyze all of the images.
I have provided some comments below that I hope will improve the manuscript.
I recommend that the introduction begin by describing the biological significance of blinking and the need for the current study. Indeed, the first two paragraphs of the discussion would be better suited in the introduction and would be a good summary of the significance.
I do not believe the authors’ predictions 2 and 3 are actually tested in the study. Both are topics for interpretation in the discussion, which is appropriate, but strictly they are not being tested with the current study design. Furthermore, the authors’ hypotheses are really research questions. There is no problem framing research in terms of questions, but it would be more appropriate to call them questions. Finally, there is no research question or prediction related to crocodilians.
I also have a few comments about the sampling regime. First, could the authors’ provide more information on how the birds (species and individuals) were selected? Was it an opportunistic sample (i.e., bird species and individuals available in the zoos, etc. you visited) or was there a more systematic approach? An opportunistic approach is fine in my view, but clarifying this would be useful for the reader. Second, was just one individual sampled per species or were multiple individuals sampled? If it was just one individual, I suggest the authors’ discuss the strength of the assumption that one or a few individuals are representative of the species as a whole.
The authors’ include many good images, but it would be beneficial if the manuscript included a graphical depiction of the proportion of species that were categorized into each of the three blink types.
In regard to figure 19, I do not see a reference to this in the text. And, there may have been an issue rendering this image on my pdf, but I found this image difficult to interpret. The figure description is too brief and the key on the image seems to be misplaced and overlapping some of the tree.
Line 48: As I am sure the authors are aware, birds are reptiles too, so 'non-avian reptile' would be more accurate here.
Line 413: A brief summary of the submergence reflex would be useful here.
Author Response
Thank you for your review which I found to be helpful. Line 39: I have moved the opening two paragraphs of the Discussion into the Introduction as suggested. Line 153: Hypotheses now include Crocodilians. Line 158: Predictions 2 and 3 have been removed. Line 223: The method of sampling is reported and the issue of observing multiple individuals of the same common species compared with a single individual of a rare species, addressed. Line 432: There is now a Table to show the proportion of each type of blink observed. Line 716: the figure has been replaced by a Table which shows the phylogeny of upper lid blinks more clearly. Line 80: 'non-avian' reptiles. Line 619: the submergence reflex is further clarified.